# Future Climate CO_2_ Reduces the Tungsten Effect in Rye Plants: A Growth and Biochemical Study

**DOI:** 10.3390/plants12101924

**Published:** 2023-05-09

**Authors:** Emad A. Alsherif, Dina Hajjar, Hamada AbdElgawad

**Affiliations:** 1Biology Department, College of Science and Arts at Khulis, University of Jeddah, Jeddah 21959, Saudi Arabia; 2Department of Biochemistry, College of Science, University of Jeddah, Jeddah 21959, Saudi Arabia; dhajjar@uj.edu.sa; 3Integrated Molecular Plant Physiology Research, Department of Biology, University of Antwerp, 2018 Antwerp, Belgium; hamada.abdelgawad@uantwerpen.be

**Keywords:** climate change, antioxidant, tocopherol, phytochelatins, redox status, tungsten

## Abstract

Heavy metal pollution is one of the major agronomic challenges. Tungsten (W) exposure leads to its accumulation in plants, which in turn reduces plant growth, inhibits photosynthesis and induces oxidative damage. In addition, the predicted increase in CO_2_ could boost plant growth under both optimal and heavy metal stress conditions. The aim of the present study was to investigate the effect of W on growth, photosynthetic parameters, oxidative stress and redox status in rye plants under ambient and elevated (eCO_2_) levels. To this end, rye plants were grown under the following conditions: ambient CO_2_ (aCO_2_, 420 ppm), elevated CO_2_ (eCO_2_, 720 ppm), W stress (350 mg kg^−1^ soil) and W+eCO_2_. W stress induced significant (*p* < 0.05) decreases in growth and photosynthesis, increases in oxidative damages (lipid peroxidation) and the antioxidant defense system, i.e., ascorbate (ASC), reduced glutathione (GSH), GSH reductase (GR), peroxidase (POX), catalase (CAT), superoxide dismutase (SOD), ASC peroxide (APX) and dehydroascorbate reductase (DHAR). On the other hand, eCO_2_ decreased W uptake and improved photosynthesis, which sequentially improved plant growth. The obtained results showed that eCO_2_ can decrease the phytotoxicity risks of W in rye plants. This positive impact of eCO_2_ on reducing the negative effects of soil W was related to their ability to enhance plant photosynthesis, which in turn provided energy and a carbon source for scavenging the reactive oxygen species (ROS) accumulation caused by soil W stress.

## 1. Introduction

Future climate change and heavy-metal-induced soil deterioration are two significant issues that require immediate attention [1,2,3]. Due to the burning of fossil fuels and shifts in land use, atmospheric carbon dioxide levels are predicted to rise globally from a pre-industrial value of roughly 280 ppm in 2020 to 550 ppm by 2050 [4]. Even today, these occurrences take place all over the globe [5]. In addition, a greater knowledge of climate change and changes in greenhouse gas concentration is necessary for crop production to be profitable in the future and to ensure sustainable and equitable food security [6]. To increase yields, increased CO_2_ is commonly used in greenhouse production of a variety of ornamental and agricultural crops [7]. According to the IPCC [8], the impact of doubling CO_2_ on various plants can result in biomass increases ranging from 10% to nearly 300%. Depending on the availability of nutrients and water, food and flower crops increase their production by an average of 30% in response to a doubling of CO_2_ [7,9], according to an analysis of hundreds of research papers.

High CO_2_ levels may have an impact on plant processes, including growth, photosynthesis, metabolite partitioning and translocation, photosynthetic enzymes, respiration rate, leaf area index, stomatal conductance, transpiration rate, biomass production and water use efficiency [10]. While investigating the effects of elevated carbon dioxide (eCO_2_), modifications to photosynthesis, biomass production and nutrient relationships were typically investigated at the physiological level in controlled environments [11,12]. To accomplish this, increasing ambient CO_2_ levels might make it easier for plants to withstand environmental heavy metal toxicity [13]. In this scenario, eCO_2_ reportedly reduced the negative effects of environmental stressors on plant growth and metabolism [14]. As an illustration, eCO_2_ decreased the detrimental impacts of various heavy metals on the metabolism and growth of plants [15,16,17]. eCO_2_ physiologically causes the plant’s metabolism to be shifted toward the production of several stress-related metabolites when additional resources (carbon) are added. In this manner, increasing dark respiration through eCO_2_ increases the accumulation and degradation of non-structural carbohydrates [18]. As a consequence, a variety of metabolites, including osmoprotectants and antioxidants, are synthesized with the help of metabolic energy [19]. eCO_2_ levels have been shown to improve plant growth and productivity by encouraging the uptake of C during photosynthetic processes and reducing photorespiration, especially in C3 plants, despite having a significant effect on the climate [11,20,21]. Additionally, Zinta et al. [21,22] discovered that eCO_2_ strengthens plants in adverse circumstances by enhancing their water use and accelerating the metabolism of their antioxidant defense.

In addition to the environmental impacts of global warming, the rapid industrial expansion has greatly increased the release of contaminants, such as heavy metals (HMs), in many ecosystems [23,24]. As a consequence, the interaction between HMs and climate change would have an impact on agriculture, which would then have an immediate effect on crop development and growth, thereby affecting production and food safety [25,26]. Changes in climate are important for studying heavy metal pollution because many environmental factors affect how heavy metals move through plants [27,28,29,30]. Shah et al. [30] claimed that HM pollution causes oxidative stress in plants by interfering with enzyme activity and substituting essential metals and nutrients, which has a negative impact on the crop quality [31].

The heavy metal tungsten (W) is employed in a variety of industrial applications [32]. Tungsten is heavily accumulated in the soil as a result of anthropogenic activity, according to soil composition [33]. Tungsten accumulation consequently affects plants and animals via the incorporation of it into the food chain. Tungsten poses a serious threat upon plant growth and development, including the retardation of seedling growth by cell cycle hindering and the perturbation of gene expression, as well as the ultrastructural malformations of cell components [34]. On the cellular and physiological levels, W was reported to impair molybdoenzymes in plants by antagonizing their Mo-Cofactors [35]. Parenthetically, W can replace molybdenum (Mo) in molybdoenzymes, thus inhibiting their catalytic activity, making them functionless, hence causing oxidative damage via triggering ROS production and inhibiting abscisic acid biosynthesis [36]. Under a certain threshold, low levels of ROS can initiate the synthesis of ROS-scavenging enzymes, meanwhile high levels of ROS causes necrosis [37]. A better knowledge of climate change and changes in greenhouse gas concentrations is necessary for agricultural production to be profitable in the future and to provide sustainable and equitable food security. Since no research has looked at how future eCO_2_ will affect tungsten-polluted soils, this research set out to explore the physiological and biochemical mechanisms of the rye plant (*Secale cereale* L.) underlying eCO_2′_s protective effect on tungsten stress for the first time.

## 2. Material and Methods

### 2.1. Plant Growth and Treatments

From the Giza Agricultural Research Center in Egypt, we procured rye seeds. The seeds were surface sterilized using sodium hypochlorite (0.5% *v*/*v*) for 20 min. After the seeds germinated on wet perlite, they were transferred into pots (20 cm tall and 15 cm in diameter) filled with 0.5 kg of organic soil and labeled as follows: pH: 7.2, organic matter: 1.1%, 57% clay, 28% silt and 15% sand. Nitrogen (25 g/g DW), P (1.6 g/g DW), K (75 g/g DW), Fe (0.84 g/g DW), Mn (1.3 g/g DW), Zn (0.1 g/g DW) and W (0.0) were the other elements. All pots received the same 3 cm of surface standing water throughout the growth period. A base fertilizer comprising 1.2 g of urea (which contains 46% nitrogen) and 1.2 g of K_2_HPO_4_ 3H_2_O (Sigma, Germany, Taufkirchen) was applied. At Jeddah University, the pots were placed in growth-controlled cabinets under precisely controlled circumstances (12 h of photoperiod, 350 mol photons m^2^ s^1^ and 28/24 °C Day and night temperatures). Before the trial began, 350 mg of tungsten (W) was added to the soil. Control solutions included soil that had not been spiked. Following the planting of the rye seeds, soils with and without spikes were developed in two different climate conditions: (1) elevated CO_2_ (eCO_2_, 720 ppm) and (2) ambient CO_2_ (aCO_2_, 420 ppm). As a result, the following conditions were used to grow the rye plants: aCO_2_ + non-spiked soil (control), eCO_2_ + non-spiked soil, aCO_2_ + spiked soil, and eCO_2_ + spiked soil. The most effective W concentration (350 mg kg^−1^ soil) was selected to reduce the growth (DW) of the delicate rye without killing the plants after a preliminary experiment with a variety of W concentrations (50–500 mg kg^−1^ soil) was completed. To keep the soil water content (SWC) at 78% throughout the trials, all pots received daily watering. To adjust for the water loss experienced by all plants (25 to 40 mL day^−1^), pots were weighted daily. Samples (shoots) were collected and stored at 80 °C for an additional biochemical study after 5 weeks of growth. Furthermore, earth samples were collected for chemical analysis.

### 2.2. Quantification of Photosynthetic Related Parameters

The stomatal permeability and light-saturated photosynthetic rate of the treated rye leaves were assessed prior to sample collection using an LI-COR LI-6400 (LI-COR Inc., Lincoln, NE, USA) [38]. Using a fluorimeter (PAM2000, Walz, Germany), the photochemical efficiency (Fv/Fm, photosystem ll system efficiency) was determined [38]. According to Hemphill and Venketeswaran [38], the concentrations of chlorophyll a and b as well as carotenoids were measured in the supernatant after the stalks were homogenized in acetone. Activities of the enzyme ribulose-1,5-bisphosphate carboxylase/oxygenase (RuBisCO) were examined by Sulpice et al. [39]. Glycerol (20%), BTriton-X100 (1%), SA (0.25%), EGTA (1 mM), MgCl_2_ (10 mM), benzamidine (1 mM), EDTA (1 mM), e-aminocapronic acid (1 mM), PMSF (1 mM), DTT (1 mM) and leupeptin (10 mM) were all chemicals from Sigma, Germany, and were used to quantify the activity.

### 2.3. Organic Acids and Phenolic Content in Soil Samples

To acquire the rhizosphere, the dug roots were progressively stirred to separate them from the bulk soils. Measurements were made of the concentrations of phenolic, oxalic and citric acids [38,39]. Organic acids (citric and oxalic acids) were gathered in 0.1% phosphoric acid that also contained butylated hydroxyanisole, using ribitol as an internal reference. Filtrates were used for HPLC quantification using a LaChrom L-7455 diode array (Merck-Hitachi, Barcelona, Spain), as described by de Sousa et al. [40], following centrifugation. According to Zhang et al. [41], the phenolic concentration was calculated using spectrophotometry (Shimadzu UV-Vis 1601 PC, Kyoto, Japan).

### 2.4. Quantification of the Tungsten in Soil and Plant

The soil samples were digested in a solution of HNO_3_-HF-HClO_4_ to determine the total soil W. The tungsten amounts in the extracted or digested solutions were measured using inductively coupled plasma mass spectrometry (ICPMS, PE, NexION 300). The powdered grasses were combined with HNO_3_-HF-HClO_4_ and the residual acids were heated at 145 °C for 2.5 h to aid in digestion. ICPMS (PE, NexION 300, Waltham, MA, USA) was used to measure the W concentration in the filtrate [42].

### 2.5. Quantification of Oxidative Damage Markers

The level of H_2_O_2_ was measured by using the FOX1 technique to observe the peroxide-mediated oxidation of Fe^2+^ and the subsequent reaction of Fe^3+^ with xylenol orange [43]. The reaction mixture containing catalase was used to evaluate the reaction specificity for H_2_O_2_ of the Fe^3+^ xylenol orange complex at a wavelength of 560 nm. The lipid peroxidation amount was extracted from rye tissues using 80% ethanol, and it was then determined using the TBA-MDA reagent [44]. By extracting lipoxygenase (LOX) in 50 mM potassium phosphate buffer (pH 7.0), 10% polyvinyl pyrrolidone (PVP), 0.25% triton X-100 and 1 mM polymethyl sulfonyl fluoride (PMSF), Steczko et al. [45] determined its activity. A microplate reader was used to quantify spectrophotometry for oxidative metabolite analyses. (Synergy Mx; BioTek Instruments Inc., Vermont, VT, USA).

### 2.6. Quantification of Antioxidant Parameters

Antioxidant concentrations and total antioxidant capacity were extracted from 200 mg FW of rye plants using 80% ethanol and centrifugation at 14,000× *g* for 18 min at 4 °C. Using a Trolox standard solution (0–650 M) and the “Ferric Reducing Antioxidant Power” assay (FRAP reagent, 0.3 M acetate buffer (pH 3.6), 0.01 mM TPTZ in 0.04 mM HCl and 0.02 M FeCl_3_·6H_2_O) [46]. By using an HPLC (Shimadzu, Hertogenbosch, the Netherlands) analysis, ascorbate (ASC) and glutathione (GSH) were identified. A total of 6% (*w*/*v*) meta-phosphoric acid was used to remove frozen plant tissue, and a reversed-phase HPLC column (100 4.6 mm Polaris C18-A, 3 m particle size, 40 °C) was used to separate the antioxidants [30]. In 80% ethanol (*v*/*v*), polyphenols and flavonoids were extracted (MagNALyser, Waregem, Belgium). The total phenolic and flavonoid content were assessed using the Folin–Ciocalteu and aluminum chloride assays, respectively [47,48]. Proteins were extracted from 200 mg FW of rye plants to determine the activity of antioxidant enzymes in two mL of a KPO_4_ extraction buffer containing polyvinylpyrrolidone (10% *w*/*v*), Triton X-100 (0.25% *v*/*v*) and phenylmethylsulfonyl fluoride at a pH of 7.0 (PMSF, 1 mM). A total of 0.05 M MES/KOH was used to spectrophotometrically assess the activities of dehydr-ASC reductase (DHAR, EC 1.8.5.1), GSH reductase (GR, EC 1.6.4.2), ASC peroxidase (APX) and monodehydro-ASC reductase (MDHAR, EC 1.6.5.4). The oxidation of pyrogallol [49] and the suppression of NBT reduction at 560 nm were used to measure the activities of peroxidase (POX, EC 1.11.1.6) and superoxide dismutase (SOD, EC 1.15.1.1) enzymes, respectively. The rates of H_2_O_2_ oxidation at 240 nm [50] and NADPH reduction at 340 nm [51] were used to measure the activities of catalase (CAT, EC 1.11.1.6) and glutathione peroxidase (GPX, EC 1.11.1.9). Enzyme activity was adjusted to the total soluble protein content using the Lowry method [52]. All metabolite and enzyme analyses were scaled down for semi–high-throughput analysis using a microplate reader (Synergy Mx; BioTek Instruments Inc., Vermont, VT, USA).

### 2.7. Quantification of Detoxification-Related Parameters

A KPO_4_ buffer (50 mM, pH 7.0) containing 0.5 mM CDNB and 1 mM GSH was used to extract glutathione-S-transferase (GST; EC 2.5.1.18) from 200 mg FW of rye plants. The activity was valued in accordance with [53]. In accordance with Diopan et al. [54], the concentration of metallothionein (MTC) was electrochemically determined using differential pulse voltammetry Brdicka reaction. After being extracted with 5% sulfosalicylic acid and combined with Ellman’s reagent, the total phytochelatins (total thiols-non-protein) were determined by spectrophotometry at 412 nm [55].

### 2.8. Statistical Analysis Experiments

Four duplicates of each treatment (*n* = 4) were used in the experiments, which used a fully randomized block design according to de Sousa et al. [41]. Levene’s and the Kolmogorov–Smirnov (SPSS)/Shapiro–Wilk (R) tests were employed to assess the homoscedasticity and normality of the data, respectively. A Tukey test following one way ANOVA (*p* < 0.05) was performed on all data (*p* < 0.05).

## 3. Results

### 3.1. Growth and Tungsten Accumulation

Treatment by tungsten only caused a significant decrease in both fresh and dry weights by 36.8% and 41%, respectively (Figure 1a,b). In contrast, the treatment with eCO_2_ only caused an insignificant increase in both the fresh and dry weights. The combined treatment (W + eCO_2_) caused an insignificant effect on both the fresh and dry weights, which indicated that the harmful effects of tungsten in fresh and dry weights were abolished when combined with eCO_2_. Rye plants that were exposed to W stress accumulated a lot of tungsten. When combined with eCO_2_, this amount dropped by 23.3% (Figure 1c).

### 3.2. Photosynthesis, Gas Exchange and Pigments

The effects of eCO_2_, W and their combination on the pigment content, photosynthesis and gas exchange in rye were detected (Figure 2). The changes in the chlorophyll content were reflected as a significant reduction in the photosynthetic rate in W-stressed plants by 47% and recorded an insignificant increase under eCO_2_ conditions (Figure 2a). The gas exchange rates (gs) exhibited an insignificant increase under eCO_2_ and W stress, while a significant increase of 28% was recorded under the combined effect of W and eCO_2_ (Figure 2b). Both chlorophyll a (Chl *a*) and chlorophyll b (Chl *b*) were significantly reduced in rye under W stress by 53.4% and 47%, respectively (Figure 2c,d). This reduction, however, was significantly attenuated when combined with eCO_2_. The carotenoid content in rye exhibited an insignificant increase under eCO_2_, and a significant increase under both treatments of V only and its combination with eCO_2_ by 90% and 150%, respectively (Figure 2e). eCO_2_ caused a significant increase in RuBisCO by 17%. In contrast, it exhibited a significant decrease under both W alone and the combination of W and eCO_2_ by 54% and 32%, respectively (Figure 2f).

### 3.3. Organic Acids and Phenolic Content in Soil

The eCO_2_ caused an insignificant increase in the phenolic compounds, citric acid and oxalic acid (Figure 3). The tungsten treatment caused an increase in the phenol, citric acid and oxalic acid of 83.2%, 180.3% and 78.4%, respectively. The combined eCO_2_ and tungsten treatment raised the phenol, citric acid and oxalic acid by 228%, 471% and 228.9%, respectively (Figure 3).

### 3.4. Quantification of Oxidative Markers

Treatments with W, eCO_2_ and their combination induced oxidative stress in rye shoots compared with their control plants (Table 1). Tungsten treatment caused a significant increase by 184.1%, 144% and 106.1% in H_2_O_2_, MDA and LOX, respectively. When W and eCO_2_ were used together, the harmful effects of tungsten were lessened by 17.7%, 35.9% and 23%, respectively. This means that eCO_2_ lessened the harmful effects of tungsten. Treatment by eCO_2_ alone caused insignificant effects on the H_2_O_2_, MDA and LOX. The treatment with tungsten alone significantly increased LOX by 106.1% (Table 1).

### 3.5. Nonenzymatic Antioxidants

The TAC activity exhibited an increase in all treatments (Table 2). Tungsten caused a significant increase in TAC by 53%, eCO_2_ caused a significant increase by only 17.7%, while the combination of W and eCO_2_ caused a significant increase by 155%. Significant increases in the phenolic content were recorded when the plants exposed to W only or V+ eCO_2_ compared to their control values, by 129% and 294%, respectively, while eCO_2_ non-significantly affected the phenolic content (Table 2). The protective role of tocopherols against W-induced stress in rye is shown in Table 2. eCO_2_, W and their combination caused significant increases in tocopherol by 20.6, 66.1% and 171.3, respectively (Table 2). eCO_2_ did not significantly affect the flavonoid content, while both W and the W + eCO_2_ caused significant increases by 307.1% and 233.9%, respectively.

### 3.6. Antioxidant Enzymes

To investigate the potential of modifications in the antioxidant defense system, we measured the concentration of antioxidant enzymes and metabolites in plants in response to the treatments. eCO_2_ showed significant increases in POX and DHAR enzymes by 36.4% and 18.1%, respectively, and had an insignificant effect on the rest of the studied antioxidant enzymes. The treatment with W only caused significant (*p* < 0.05) increases for GR (15%), GPX (134%), POX (105.8%), CAT (148.4%), SOD (135%), APX (135.7%), DHAR (119.6%), ASC (104.4%) and GSH (161.9%) (Table 3). The effect of the combined treatment with W and eCO2 showed the highest increases for the studied antioxidant defense system as follows: 301.4% of GR, 237.3% of GPX, 238.6% of POX, 142.3% of CAT, 124% of SOD, 242.8% of APX, 169.6% of DHAR, 97.7% of ASC and 300% of GSH.

### 3.7. Detoxification Metabolties and Enzymes

Figure 4 shows the changes in metallothioneins (MTC), phytochelatins (PC), Tgsh and GST activities in rye plants exposed to eCO_2_, W or their combination. Under eCO_2_ treatment, the measured parameters exhibited insignificant increases compared to their control counterparts.

Treatment by W only caused significant increases by 61.3%, 160%, 188.8% and 98.5% for PC, Tgsh, GST and MTC, respectively. The combined treatment showed significant increases by 166.3% for PC, 280% for Tgsh, 300% for GST and 161.3% for MTC.

## 4. Discussion

In comparison to untreated controls, rye plant growth reductions were more pronounced in FW and DW under W stress (a 36.8% and 41.8% drop, respectively). Adamakis et al. [34] reported that the first observable sign of W toxicity was typically a significant decrease in the development of plant shoots and roots. Similar findings were recorded by Preiner et al. [36] and Adamakis et al. [56], who discovered that *Glycine max*, *Pisum sativum* and *Gossypium hirsutum* all experienced considerably less growth as the W concentration increased. The growth of other species such as *Helianthus annuus* and *Brassica oleracea* was also significantly reduced in reaction to W contamination [57]. Several hypotheses have been put forth to explain why W has a negative impact on plant development. These hypotheses included W and Mo’s shared chemical properties as necessary plant micronutrients, whereas controlling N assimilation by W interferes with molybdoenzyme activity, which is essential for plant development and adaptation to environmental challenges [36,58]. The adverse impact of W on photosynthetic efficiency may also be linked to the decrease in biomass in the tested crops. As expected, W caused a significant decrease in the photosynthetic efficiency in the target plants, which in turn caused a delay in plant development (Figure 2). In the present study, Figure 2a shows that the rye photosynthetic rate experienced a significant slowdown due to W stress (47.9%). The substantial drop (*p* ˂ 0.05) in Chl *a* and *b* levels, as well as in stomatal conductance and RuBisCO activity, and the crops grown in W stress conditions, may be used to explain this obstruction in photosynthesis. On the other hand, plants treated with W had substantially higher levels of carotenoids (Figure 2e). Johnson et al. [57] and Kumar and Aery [35] stated that the W therapy decreased Chl *a* and/or Chl *b* while significantly increasing the carotenoids in several species, including wheat and *Helianthus annuus*. It is noteworthy that heavy metals can have various effects on the photosynthesis apparatus. They may build up in plant leaves, separate in leaf tissues, interact with important photosynthesis enzymes, change chloroplast membranes and then engage in interactions with photosystems [58]. Tungsten, on the other hand, may indirectly influence plant development by changing the pH of the soil and interfering with the soil’s ability to retain nutrients. One such example is the oxidation of W in soil to tungstate, which can then be polymerized with phosphate [59]. As a result, it causes phosphate deficiency, which negatively impacts photosynthesis, respiration, glycolysis and ATP production [36].

Data indicated that stressed plants had considerably higher levels of H_2_O_2_. In general, oxidative stress results from an imbalance between pro-oxidant and antioxidant levels, which disrupts the equilibrium of antioxidants [37]. Similar to this, oxidative disturbances brought on by heavy metal pollution can throw homeostasis out of balance. Consequently, oxidative stress negatively impacts the cellular structure, causing unspecific protein, DNA and lipid (MDA) breakdown [60,61]. The concentrations of MDA and the activity of LOX provide a reasonable representation of the oxidative status of plants and are one of the most significant stress-induced distractions [62,63].

Based on what we found, this rise in H_2_O_2_ levels happened at the same time that MDA levels and lipoxygenase activity also went up (Table 1). According to Xu et al. [64], W significantly raised the amounts of H_2_O_2_ in *Solanum nigrum*. Due to disturbances in the oxidative balance, the cell also makes more ROS such as hydroxide radicals (HO), superoxide anions (O_2_), and [65]. This could very well explain why the H_2_O_2_ levels in all stressed rye were higher than in their normal control crops. According to the analysis of non-enzymatic antioxidants, we also significantly raised the levels of polyphenols, tocopherols (which are indicative of antioxidative membrane protection) and flavonoids in all of the tested crops, but these increases were insufficient to fully mitigate oxidative damage (Table 2). Our results are consistent with those of Saleh et al. [15], who discovered that Triticum aestivum cultivated in NiO-NPs-contaminated soil had a marginally higher TAC. Similarly, corn and barley both had significantly higher TAC, flavonoids and polyphenol levels after exposure to arsenate [19]. Furthermore, yellow lupin and Panax ginseng that have been treated with Pb or Cu have more phenolic substances, especially flavonoids [66]. When the levels of non-enzymatic antioxidants such as the total antioxidant capacity, polyphenols, tocopherols and glutathione (GSH) go up, they cause reactive oxygen species (ROS) to be made, which these non-enzymatic antioxidants then get rid of [67]. These included peroxidase (POX), superoxide dismutase (SOD), catalase (CAT), glutathione reductase (GR) and ascorbate peroxidase (APX). In the current results, the POX activity in rye increased by 105% when it was treated with W. We looked at how the detoxification system in rye responded to eCO_2_ under control and W stress circumstances in order to conduct further research on the ameliorative effects of eCO_2_ on plants under the stress of W. Under W treatment circumstances, metallothioneins (MTC), phytochelatins (PCs) and GSH-S-transferase (GST) levels in rye significantly increased (Figure 4).

The impacts of heavy metal stress are mitigated by eCO_2_ levels, according to earlier research [68,69], but it is unknown whether this is true for W. Here, we demonstrated that eCO_2_ did, in fact, lessen the W toxicity, demonstrating that eCO_2_ functions as an efficient and effective method that can lessen W toxicity in rye plants. Numerous studies have revealed that eCO_2_ can reduce the stress caused by heavy metals [70,71]. Here, we demonstrated that W is also affected by this. Increased CO_2_ can provide additional C while also causing stomatal closure or lowering non-stomatal variables such as ROS production and photorespiration [72,73]. In the current research, eCO_2_ significantly decreased in photorespiration, including the HDR, GO and G/S ratio, caused by Cr. Prior research by Zinta et al. [21] demonstrated that the presence of eCO_2_ encourages carboxylation over oxygenation at RuBisCO, thereby decreasing the production of ROS. Importantly, eCO_2_ can maintain the energy needed for stressed plant development as well as the C skeletons [74]. The supply of antioxidant molecules may grow as the C availability rises, improving defenses against oxidative damage [75]. We observed a significant rise in the TAC, indicating that plants up their antioxidant defenses to combat the ROS production brought on by Cr toxicity. Under W stress, we also observed a rise in tocopherol, phenol, ASC, flavonoids, GSH and PC. According to Di Toppi et al. [72], under arsenic stress, the antioxidant defense system of tomato is activated, which is consistent with our findings [13]. In contrast to ambient CO_2_, the antioxidant levels are greater under eCO_2_ conditions. Plants generate a large number of enzymatic molecules in addition to enzymatic antioxidants. According to the current research, many enzymes were significantly induced by eCO_2_ under W stress as opposed to aCO_2_, indicating that eCO_2_ is responsible for the rice plants’ increased antioxidant defense system; by increasing antioxidant enzymes, W poisoning is reduced [76]. According to Bencze et al. [75], wheat’s antioxidant enzymes were boosted by eCO_2_ conditions under drought stress. In chamomile plants, SOD was activated when exposed to Cr [77], which is consistent with our findings. In the present research, tungsten can be less dangerous by increasing the release of oxalic and citric acids into the soil through the roots. It was documented that when plant roots are stressed, they release more organic acids into the soil, which combine with chelates to change how heavy metals are held in place and moved around in the soil [78]. It is interesting to note that rye plants’ oxidative damage from W was reduced by eCO_2_. Several studies [79,80] have previously reported on this mitigation effect. Increased CO_2_ has the potential to add more C for the synthesis of antioxidants. It is significant that eCO_2_ can keep stressed plants’ C skeletons and energy levels stable [43]. To improve protection against oxidative damage, AbdElgawad et al. [70] found that increasing the availability of carbon dioxide led to a rise in the production of antioxidant molecules. Rye’s antioxidant enzymes were enhanced by eCO_2_ settings under environmental stress [80]. The W exposure increases APX, GPX, CAT, SOD, DHAR and GR activities, though more so in plants subjected to eCO_2_ levels. In addition to increasing the generation of antioxidants, eCO_2_ decreased photorespiration and ROS [74,75]. According to the current study, eCO_2_ significantly reduced W-induced increases in photorespiration, including HDR and Gr. Prior studies by Zinta et al. [21] showed that eCO_2_ improved the carboxylation over oxygenation of RuBisCO enzymes, thereby reducing the ROS production.

All of this clarified how eCO_2_ can lessen the phytotoxicity risks of W in rye plants. Our research demonstrates that the positive impact of eCO_2_ on reducing the negative effects of soil W was related to their ability to enhance plant photosynthesis, which in turn provided energy and a carbon source for scavenging the ROS accumulation caused by soil W stress.

## 5. Conclusions

This study advances our knowledge of the processes behind the variations in the physiological and biochemical responses of the rye crop under W and eCO_2_ conditions. According to this study, eCO_2_ reduces the effects of W stress by elevating antioxidant levels and the activity of antioxidant enzymes, especially in areas with soil and conditions similar to the study conditions.

## Figures and Tables

**Figure 1 plants-12-01924-f001:**
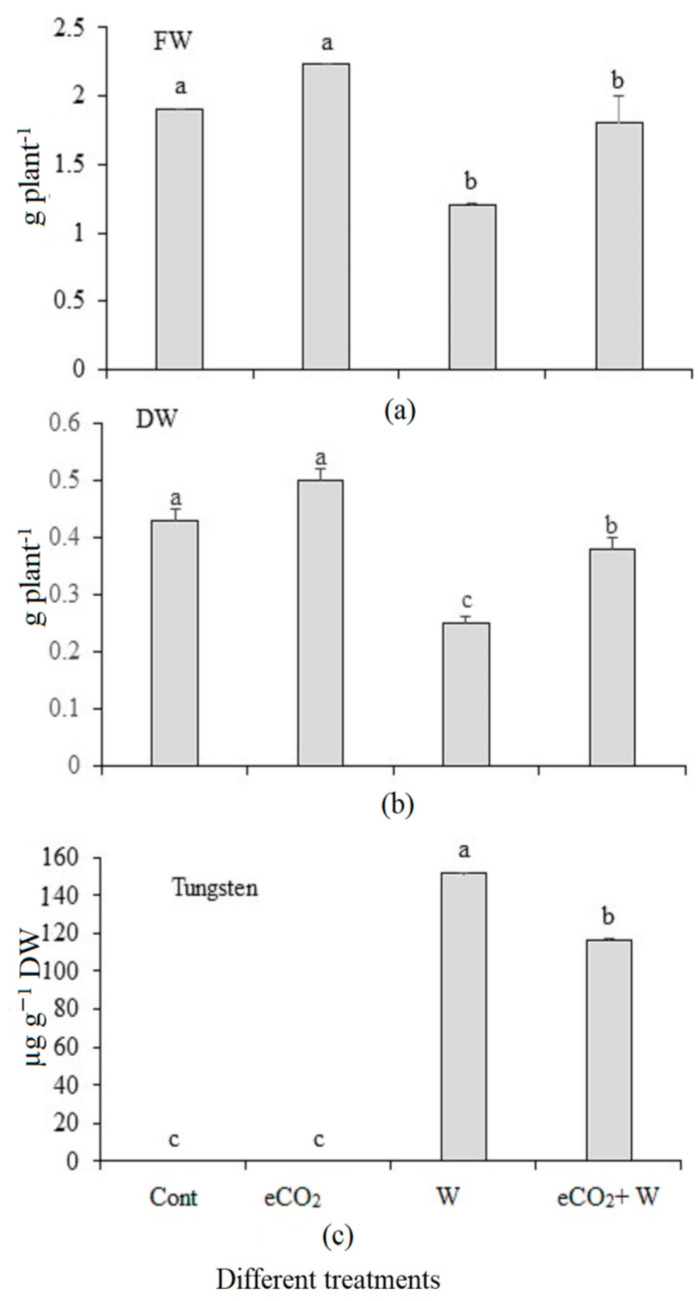
Effect of elevated CO_2_, tungsten (W) and their combination on vanadium concentration, fresh weight and dry weight of rye. Cont.: ambient CO_2_ (410 ppm); eCO_2_: (620 ppm), V: 350 mg kg^−1^ soil. The aforementioned information is presented as mean values with standard error (*n* = 4). Different letters indicate significantly different means in the Tukey test following one way ANOVA (*p* < 0.05). Different letters denote statistically significant differences between the means of the same plant species, at least at the 0.05 level of significance.

**Figure 2 plants-12-01924-f002:**
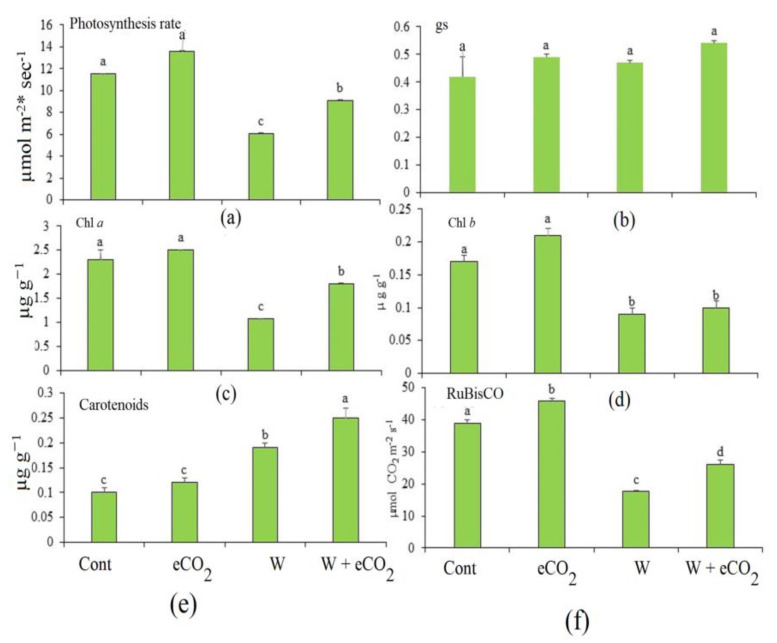
Effect of elevated CO_2_ (eCO_2_), tungsten (W) and their combination on photosynthesis (**a**) mol CO_2_/m^2^/S), (**b**) stomatal conductance (gs) (nmol CO_2_ m^−2^ s ^−1^), (**c**) Chl *a* (mg/gFW), (**d**) Chl *b* (mg/g FW), (**e**) carotenoids (mg/g FW) (D) and (**f**) RuBisCO (nmol 3-PGA/mg protein min.) of rye. Data are mean values ± SE (*n* = 4). Different letters indicate significantly different means in the Tukey test following one way ANOVA (*p* < 0.05). Different letters indicate that there is a significant difference between the treatments.

**Figure 3 plants-12-01924-f003:**
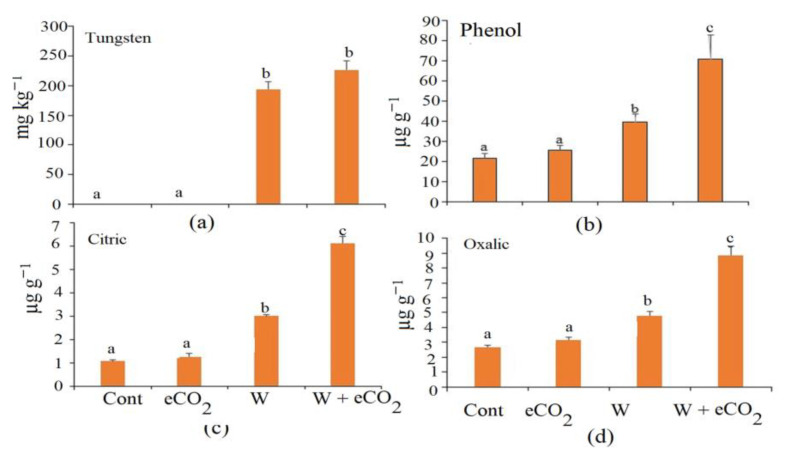
Effect of elevated CO_2_ (eCO_2_), tungsten (W) and their combination on tungsten, polyphenol, citric acid and oxalic acid in soil. Data are mean values ± SE (*n* = 4). Different letters indicate significantly different means in the Tukey test following one way ANOVA (*p* < 0.05). Different letters indicate that there is a significant difference between the treatments.

**Figure 4 plants-12-01924-f004:**
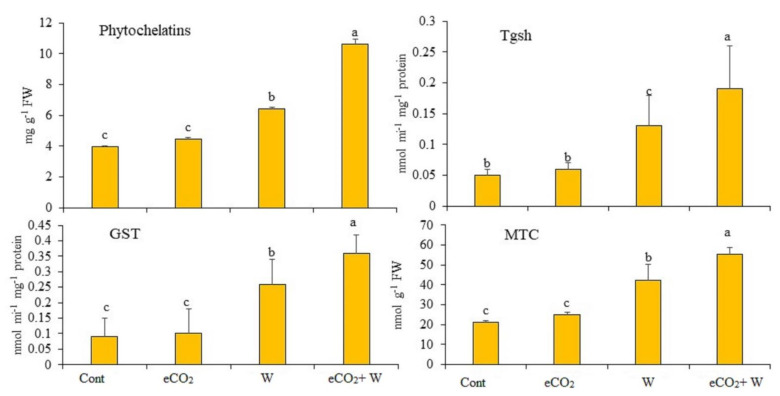
Effect of elevated CO_2_ (eCO_2_), tungsten (W) and their combination on phytochelatins, total glutathione (Tgsh), and glutathione-S-transferase (GST) and metallothioneins (MTC). Data are mean values ± SE (*n* = 4). Different letters indicate significantly different means in the Tukey test following one way ANOVA (*p* < 0.05).

**Table 1 plants-12-01924-t001:** Effect of elevated CO_2_, tungsten (W) and their combination on oxidative markers.

	H_2_O_2_	MDA	LOX
	N mol g^−1^ FW		
Control (ambient CO_2_—410 ppm)	291 ± 5.7 ^c^	3.34 ± 0.06 ^c^	1.94 ± 0.04 ^c^
eCO_2_ (620 ppm)	308 ± 19.0 ^c^	3.14 ± 0.08 ^c^	1.79 ± 0.15 ^c^
W (350 mg kg^−1^)	827 ± 14.0 ^a^	8.15 ± 0.38 ^a^	4 ± 0.06 ^a^
eCO_2_ + W	680 ± 12.0 ^b^	5.22 ± 0.11 ^b^	3.08 ± 0.10 ^b^

H_2_O_2_: hydrogen peroxide, MDA: malondialdehyde and LOX: lipoxygenase. Cont.: ambient CO_2_ (410 ppm); eCO_2_: (620 ppm), V:350 mg kg^−1^ soil. The aforementioned information is presented as mean values with standard error (*n* = 4). One way ANOVA and the Tukey post hoc test were used to statistically examine the data and compare the means. Different letters denote statistically significant differences between the means of the same plant species, at least at the 0.05 level of significance.

**Table 2 plants-12-01924-t002:** Effect of elevated CO_2_ (eCO_2_), tungsten (W) and their combination (eCO_2_ + W) on non-enzymatic antioxidants of rye.

	TAC	Pphenol	Flav	Ttoco
	mg g^−1^ FW			
Control (ambient CO_2_—410 ppm)	34.30 ± 0.9 ^d^	1.22 ± 0.04 ^c^	0.56 ± 0.01 ^a^	21.3 ± 0.43 ^c^
eCO_2_ (620 ppm)	40.39 ± 1.3 ^c^	1.44 ± 0.06 ^c^	0.66 ± 0.01 ^a^	25.7 ± 0.92 ^c^
W (350 mg kg^−1^)	52.87 ± 0.8 ^b^	2.80 ± 0.07 ^b^	2.28 ± 0.06 ^b^	35.4 ± 0.96 ^b^
eCO_2_ + W	87.55 ± 2.4 ^a^	4.81 ± 0.40 ^a^	1.87 ± 0.02 ^a^	57.8 ± 1.34 ^a^

TAC: total antioxidant capacity, Pphenol: Polyphenol, Flav: flavonoid and Ttoco: total tocopherol. Data are mean values ± SE (*n* = 4). Data are mean values ± SE (*n* = 4). Different letters indicate significantly different means in Tukey test following one way ANOVA (*p* < 0.05).

**Table 3 plants-12-01924-t003:** Effect of elevated CO_2_ (eCO_2_), tungsten (W) and their combination (eCO_2_ + W) on antioxidant parameters of rye.

	GR	GPX	POX	CAT	SOD	APX	DHAR	ASC	GSH
	N mol min^−1^ mg^−1^ Protein							mg g^−1^ FW	
Control (ambient CO_2_—410 ppm)	0.070 ± 0.01 ^c^	0.142 ± 0.012 ^a^	0.546 ± 0.03 ^d^	3.703 ± 0.104 ^b^	100 ± 2.2 ^b^	0.14 ± 0.01 ^c^	0.066 ± 0.003 ^b^	0.089 ± 0.002 ^b^	0.021 ± 0 ^c^
eCO_2_ (620 ppm)	0.082 ± 0.01 ^c^	0.185 ± 0.01 ^a^	0.743 ± 0.01 ^c^	4.126 ± 0.24 ^b^	118 ± 11.2 ^b^	0.16 ± 0.02 ^c^	0.078 ± 0.01 ^a^	0.105 ± 0.010 ^b^	0.024 ± 0.0 ^c^
W (350 mg kg^−1^)	0.150 ± 0.01 ^b^	0.334 ± 0.03 ^b^	1.124 ± 0.06 ^b^	9.199 ± 0.13 ^a^	253 ± 2.7 ^a^	0.33 ± 0.04 ^b^	0.145 ± 0.01 ^a^	0.182 ± 0.002 ^a^	0.055 ± 0.0 ^b^
eCO_2_ + W	0.281 ± 0.01 ^a^	0.479 ± 0.01 ^a^	1.849 ± 0.10 ^a^	8.973 ± 0.16 ^a^	224 ± 10.0 ^a^	0.48 ± 0.02 ^a^	0.178 ± 0.01 ^a^	0.176 ± 0.001 ^a^	0.084 ± 0.0 ^a^

GR: Glutathione reductase, GPX: peroxidase, POX: peroxidase, CAT: catalase, SOD: superoxide dismutase, APX: ascorbate peroxidase DHAR: dehydroascorbate reductase, ascorbate (ASC), and reduced glutathione (GSH) and peroxidase (POX). Data are mean values ± SE (*n* = 4). Different letters indicate significantly different means in the Tukey test following one way ANOVA (*p* < 0.05).

## Data Availability

The data presented in this study are available on request from the corresponding author.

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
