# Peer review of "Future Climate CO2 Reduces the Tungsten Effect in Rye Plants: A Growth and Biochemical Study"

_plants, 2023, doi:10.3390/plants12101924_

Round 1

Reviewer 1 Report

Dear Authors and Editors

The article „Future climate CO2 reduces the Tungsten effect in rye plants: A

growth and biochemical study” was written in a transparent manner. In its assumptions, the study is interesting.

The topics presented in the article are appropriate for the Journal's profile.

The authors presented the discussed issues in a broad way.

The literature is selected in the right way.

The conclusions sum up the entire article appropriately.

However, I have a few technical notes:

- - page 3 point 2.1. - Space after K2HPO4;

- page 3 point 2.4. - In the formula MgCl2, "2" should be written in subscript;

- page 4 point 2.6. - correct formula FeCl3.6H2O; - In the KPO4 formula, "4" should be entered in subscript;

- page 7 in the description to Fig 2 and in point 3.3. - in the formula CO2, "2" in subscript;

- in the discussion section, in CO2 formulas, "2" is not always in subscript;

- Please standardize references;

- in Table 1, 2, 3 - the presented values should have the same number of significant digits,

In my opinion, this paper can be accepted for publication in Plants after minor corrects.

Best regards

Author Response

Dear Authors and Editors

The article „Future climate CO2 reduces the Tungsten effect in rye plants: A

growth and biochemical study” was written in a transparent manner. In its assumptions, the study is interesting.

The topics presented in the article are appropriate for the Journal's profile.

The authors presented the discussed issues in a broad way.

Literature is selected in the right way.

The conclusions sum up the entire article appropriately.

However, I have a few technical notes:

Comment - page 3 point 2.1. - Space after K2HPO4;

Response: We added the space

Comment - page 3 point 2.4. - In the formula MgCl2, "2" should be written in subscript;

Response: It is written in subscript

Comment - page 4 point 2.6. - correct formula FeCl3.6H2O; - In the KPO4 formula, "4" should be entered in subscript;

Response: They are written in subscript

Comment - page 7 in the description to Fig 2 and in point 3.3. - in the formula CO2, "2" in subscript;

Response: It is written in subscript

Comment - in the discussion section, in CO2 formulas, "2" is not always in subscript;

Response: We revised al CO2 formulas and corrected it

Comment - Please standardize references;

Response:

Comment - in Table 1, 2, 3 - the presented values should have the same number of significant digits,

Response: The presented values were written with the same significant digits

In my opinion, this paper can be accepted for publication in Plants after minor corrects.

Best regards.

Comment: Thank you

Reviewer 2 Report

Dear authors,

This paper is very interesting combining the effect of eCO2 with tungsten on rye plants. although it is well written , it should be reorganized and some other comments should be considered as follows:

Why you speak about heavy metals in the first

The abstract should be extended to include the treatments and methods used

Don't start the sentence with abbreviations such as eCO2, W,….

Please spell out the ROS for the first time, initially in the keyword

The sentence "sodium hypochlorite surface cleaning (0.5% v/v; 20 min)." is not a complete sentence

In section 2.1: what do mean by 28% sediment is this silt or you mean that you mixed a sediment with the soil because even a sediment (the soil in water environments) contents from clay, sand , and silt also

Change the units to SI units such as % to g kg-1 , g/g to g g-1 and so on through the whole manuscript

In page 3, you wrote "Following the planting of wheat seeds"although you mentioned above and blow rey plants

350 Na3VO3 mg of vanadium is this a source for W please clarify

Why you determined 78% to keep water in the soil

Please write the manufacturing country of ICP

The word broken down in the first sentence in section 2.3. is not suitable here, so please replace it

In section 2.6 and 2.7, you didn't mentioned the target of the analysis is this soil or plant and if it is plant, what its weight and is it wet or dry…

The results should presented as the same order in the analysis, I mean you started  the analysis with organic acids then tungsten content …, however in the results the order is totally different

Please replace the word non-significant with insignificant in the result section

Fig1: the word vanadium is confusing if it is a source for tungsten, it is better to write tungsten

Section 3.7 in the results is displayed differently that the analysis "Quantification of detoxification related parameters" , this will confuse the reader

The place of fig3 is replaced with fig4

Is this sentence written correct: In the current research, eCO2 significantly decreased increases in photorespiration

In the sentence "According to di Toppi et al. [77] and arsenic stress, antioxidant defense system is activated, which is consistent with our findings", the arsenic may differ than tungsten in its properties, availability and so on, so the similarity is not suitable here

In the conclusion, you generalize the results while your soil is differ than other soils under the same conditions , so please rewrite the conclusion to clarify your results and its importance to the specific field , the obstacles, and future interests

The tiles of the tables should be short and indicative, while the abbreviations should be written blow the table. 

The quality of English language is well to some extent, however some sentences should be rephrased 

Author Response

Dear authors,

This paper is very interesting combining the effect of eCO2 with tungsten on rye plants. although it is well written, it should be reorganized and some other comments should be considered as follows:

Comment: Why you speak about heavy metals in the first

Response: Because tungsten is a heavy metal.

Comment: The abstract should be extended to include the treatments and methods used

Response: Thank you for your comment, but unfortunately, because the abstract should be a total of about 200 words maximum (as written in the instructions to authors), we tried to write the most important. The present abstract is 227 words long.

Comment: Don't start the sentence with abbreviations such as eCO2, W,….

Response: We revised all manuscript and found four sentences start with W and replaced it by Tungsten.

Comment: Please spell out the ROS for the first time, initially in the keyword

Response: Wespelled it in the kewords (Reactive oxygen species).

Comment: The sentence "sodium hypochlorite surface cleaning (0.5% v/v; 20 min)." is not a complete sentence

Response: We changed it to “. The seeds were surface sterilized using sodium hypochlorite (0.5% v/v) for 20 minutes.”.

Comment: In section 2.1: what do mean by 28% sediment is this silt or you mean that you mixed a sediment with the soil because even a sediment (the soil in water environments) contents from clay, sand, and silt also

Response: Thank you for observing this error, we meant 28% sand.

Comment: Change the units to SI units such as % to g kg-1, g/g to g g-1 and so on through the whole manuscript

Response: Thank you for your comment, but we believed that the current units is easy to the reader.

Comment: In page 3, you wrote "Following the planting of wheat seeds"although you mentioned above and below rye plants

 Response: We are sorry for that error; it is rye NOT wheat. We corrected it

Comment: 350 Na3VO3 mg of vanadium is this a source for W please clarify

Response:

Comment; Why you determined 78% to keep water in the soil

Response: Because 70-80% of SWC is the optimum condition for best growth (No drought and flooding).

Comment: Please write the manufacturing country of ICP

Response: We write it (Waltham, USA)

Comment: The word broken down in the first sentence in section 2.3. is not suitable here, so please replace it

Response: We replaced it by (digested in)

Comment: In section 2.6 and 2.7, you didn't mentioned the target of the analysis is this soil or plant and if it is plant, what its weight and is it wet or dry…

Response: Thanks, these antioxidants were extracted form plants (0.2g FW), these details were added.

Comment: The results should presented as the same order in the analysis, I mean you started the analysis with organic acids then tungsten content …, however in the results the order is totally different

Response: we rearranged the analysis in section Materials and Methods as it in the Result section

Comment: Please replace the word non-significant with insignificant in the result section

Response: We replaced all (non-significant) to insignificant

Comment Fig1: the word vanadium is confusing if it is a source for tungsten, it is better to write tungsten.

Response: Sorry for this error and thank you for your observation, we changed it to tungsten.

Comment Section 3.7 in the results is displayed differently that the analysis "Quantification of detoxification related parameters”, this will confuse the reader

Response: Thanks, it is rephrased for clarity.

Comment The place of fig3 is replaced with fig4.

Response: Thank you for your observation and we are sorry for this error, we replaced it.

Comment Is this sentence written correct: In the current research, eCO2 significantly decreased increases in photorespiration.

Response: The sentence is not correct, so we removed the word increase

Comment In the sentence "According to di Toppi et al. [77] and arsenic stress, antioxidant defense system is activated, which is consistent with our findings", the arsenic may differ than tungsten in its properties, availability and so on, so the similarity is not suitable here

Response: We add this reference as the arsenic is a heavy metal like tungsten.

Comment In the conclusion, you generalize the results while your soil is differed than other soils under the same conditions, so please rewrite the conclusion to clarify your results and its importance to the specific field, the obstacles, and future interests

Response: Thank you for your comment. We add the sentence (especially in areas with soil and conditions similar to the study conditions) to remove generalization.

Comment The titles of the tables should be short and indicative, while the abbreviations should be written blow the table. 

Response: We shortened the table titles and put the abbreviations below the table.

Reviewer 3 Report

In this study, authors investigated the effect of W on the growth, photosynthetic parameters, oxidative stress, and redox status in rye plants under ambient and elevated (eCO2) levels. Here are some suggestions:

1.        In the section of “Materials and methods”, authors referred that 250 mg/Kg soil was the most effective W concentration based on their preliminary experiment. Then why authors added 350 mg of W into the soil in this experiment?

2.        I suggest authors to provide the photos of the plant growth phenotype.

3.        The title of the y-axis should be provided in each figure.

4.        Why did authors use Tukey posthoc test to analyze the significance in some figures, while they use Fisher’s LSD test in other figures? The significance indicated by different letters need to be checked. I do not think it meets a criterion.

5.        In the section of “Introduction”, authors should elucidate more clearly about the current status of W pollution in the soil worldwide, and how W pollution affected the crop growth and grain yield?

Author Response

In this study, authors investigated the effect of W on the growth, photosynthetic parameters, oxidative stress, and redox status in rye plants under ambient and elevated (eCO2) levels. Here are some suggestions:

  1. In the section of “Materials and methods”, authors referred that 250 mg/Kg soil was the most effective W concentration based on their preliminary experiment. Then why authors added 350 mg of W into the soil in this experiment?

Response: Thanks for valuable observation, 350mg/kg soil is used. This is corrected across the manuscript.

  1. I suggest authors provide the photos of the plant growth phenotype.

Response: Unfortunately, we have not photos of the plant growth phenotype

  1. The title of the y-axis should be provided in each figure.

Response: Thank you for your comment. The titles of all axes are the same and provided below the figure in the figure captions.

  1. Why did authors use Tukey posthoc test to analyze the significance in some figures, while they use Fisher’s LSD test in other figures? The significance indicated by different letters need to be checked. I do not think it meets a criterion.

Response: Thanks for valuable observation, Different letters indicate significantly different means in Tukey test following One Way ANOVA (P<0.05). This is corrected across the manuscript.

  1. In the section of “Introduction”, authors should elucidate more clearly about the current status of W pollution in the soil worldwide, and how W pollution affected the crop growth and grain yield?

Response: Thank you for your variable comment, we added the current status of W pollution in the last paragraph.

Round 2

Reviewer 2 Report

Dear authors,

The paper has been improved, however some comments are ignored and should be considered. 

Some of the method used should be clarified in the abstract and the abstract already are more than the required words, so you can write the most important sentences including: background and aim, methods, results and the importance of the findings to the field

Comment: Please spell out the ROS for the first time, initially in the keyword

Response: We spelled it in the keywords (Reactive oxygen species): the first mention is in the abstract and even the spelled words in the keyword were written without abbreviation

Comment: In section 2.1: what do mean by 28% sediment is this silt or you mean that you mixed a sediment with the soil because even a sediment (the soil in water environments) contents from clay, sand, and silt also

Response: Thank you for observing this error, we meant 28% sand.

In the response you wrote sand and in the manuscript you wrote silt which one is correct

Comment: Change the units to SI units such as % to g kg-1, g/g to g g-1 and so on through the whole manuscript

Response: Thank you for your comment, but we believed that the current units is easy to the reader.

The units supported now are the international units (SI), so the units should be changed into SI

Author Response

The paper has been improved; however some comments are ignored and should be considered. 

Comment: Some of the method used should be clarified in the abstract and the abstract already are more than the required words, so you can write the most important sentences including: background and aim, methods, results and the importance of the findings to the field

Response: We rewrite the abstract as the reviewer request.

Comment: Please spell out the ROS for the first time, initially in the keyword.

Response: We spelled it in the keywords (Reactive oxygen species):

Comment: The first mention is in the abstract and even the spelled words in the keyword were written without abbreviation.

Response: We wrote the abbreviation in the abstract where the first mention.

Comment: In section 2.1: what do mean by 28% sediment is this silt or you mean that you mixed a sediment with the soil because even a sediment (the soil in water environments) contents from clay, sand, and silt also. In the response you wrote sand, and, in the manuscript, you wrote silt which one is correct.

Response: We confirm that is 28 % silt as written in the manuscript.

Comment: Change the units to SI units such as % to g kg-1, g/g to g g-1 and so on through the whole manuscript

Response: We changed all the units as SI units, we wrote % in the manuscript to show the amount/ quantity of change NOT as units.